# Turnip Yellows Virus Field Infection in Oilseed Rape: Does It Impact the Yield and Quality?

Ramóna Vizi, József Kiss *, György Turóczi, Nóra Dobra and Zoltán Pálinkás

Department of Integrated Plant Protection, Institute of Plant Protection, Hungarian University of Agriculture and Life Sciences, H-2100 Gödöllő, Hungary; vizi.ramona@gmail.com (R.V.); turoczi.gyorgy@uni-mate.hu (G.T.); dobranora27@gmail.com (N.D.); palinkas.zoltan@uni-mate.hu (Z.P.)
* Correspondence: jozsef.kiss@uni-mate.hu

**Abstract:** *Brassica napus* L., winter oilseed rape (OSR), is a major crop worldwide, with a wide range of uses and high profitability. Viruses, such as turnip yellows virus (TuYV), are becoming increasingly important, and in certain years, they can cause significant infestations in OSR. It is difficult to detect the presence of the virus during visual field inspections, as the symptoms it causes can be confused with either those caused by abiotic factors (e.g., low winter temperature, soil compaction, nutrient deficiencies, etc.) or by other viruses. The objective of this study was to determine the susceptibility of four commercial hybrids of oilseed rape to TuYV in Hungary and to determine the effect of the virus on phenotypic and yield parameters. The results showed that infection with the TuYV can be significant in OSR in some growing seasons. It was found that the appearance of visual symptoms (e.g., anthocyanin leaves) does not always confirm the presence of the virus (based on the ELISA (enzyme-linked immuno assay)), and it does not always detect a negative effect of TuYV on all phenotypic and yield parameters of the tested hybrids based on the results in one growing season.

**Keywords:** *Brassica napus* L.; viral disease; TuYV; occurrence; yield losses; economic importance

## 1. Introduction

*Brassica napus* L., winter oilseed rape (OSR), is an important crop in the European Union, including Hungary. About 5.3 million hectares are cultivated in the European Union, of which 258 thousand hectares are dedicated to OSR cultivation in Hungary [1]. OSR is cultivated primarily for its high oil content and as a significant raw material for biodiesel production and the food industry. OSR cultivation (as a component of arable cropping systems) significantly impacts soil structure and organic matter content, which is why it may also be successfully grown as a green manure crop [2]. In addition, OSR cultivation delivers many advantages, as it may be easily integrated into arable cropping systems as well as being a profitable crop. However, several pathogens may reduce the yield quantity and quality. A wide range of fungal pathogens (*Sclerotinia sclerotiorum* (Lib.) de Bary; *Alternaria brassicae* (Berk.) Sacc.; *A. brassicicola* (Schwein.) Wilt.; *Plenodomus lingam* (Tode ex Fr.) Höhn [3,4]) are major yield-limiting factors in OSR cultivation. Some viruses (cauliflower mosaic virus (CaMV), turnip yellows virus (TuYV), and turnip mosaic virus (TuMV)) may also reduce the yield [5].

The TuYV has been detected in crops in several European countries, including France, Germany, Poland, Ukraine, Austria, Denmark, Czech Republic, Greece, and Serbia [6–10]. Moreover, it is also present in South Africa, India, China, and Australasia [11–14].

The TuYV has a wide host range, including many crop species (e.g., OSR, Indian mustard, cauliflower, and many other species of *Brassica*) and weeds from more than twenty plant families (e.g., *Amaranthaceae* (*Amaranthus retroflexus*), *Asteraceae* (*Taraxacum officinale*), *Brassicaceae* (*Brassica napus, Capsella bursa–pastoris, Descurainia sophia, Raphanus raphanistrum, Sinapis alba*)) [15,16]. TuYV was detected in the pod components of OSR seeds (ELISA test), as it was present in the pod wall, septum, and seed coat; however, it was present in

only 2 out of 78 embryos [15,17]. TuYV cannot be transmitted mechanically or with seed, but different aphid species are able to transmit it persistently, which does not pass on to their offspring [15,17,18]. Many aphid species are vectors of TuYV, such as *Brachycorynella asparagi, Cavariella aegopodii, Macrosiphoniella sanborni, Macrosiphum albifrons, Myzus nicotianae, Nasonovia ribisnigri, Pentatrichopus fragaefolii, Rhopalosiphum maidis* and *Sitobion avenae* [18]. Among the TuYV aphid vectors, *M. persicae* proved to be the most effective, with a transmission rate of 96.4%, while *M. euphorbiae* showed the lowest rate, with only 3.5%. In Hungary, the optimal time for sowing winter OSR is the latter part of August [19], while the main flight period of *M. persicae* species is October–November. As a result, the transmission of TuYV and the infection of the OSR plant with the virus mainly occurs at the beginning of the growing season [8,20].

OSR infected with TuYV reveals a wide range of symptoms in different host plants. Some host plants infected with TuYV show dwarfing, anthocyanin discolouration on leaf margins, and interveinal yellowing or reddening. In addition, TuYV symptoms may include curling, thickening, and brittleness of leaves or stunting of plants. The most typical symptom in the case of OSR is the reddish discolouration appearing on the older leaves of the plants, which may spread to the whole plant [15]. These symptoms are difficult to distinguish from those caused by other abiotic factors (e.g., soil compaction, nutrient deficiency, water stress, frost damage, or even natural senescence) through the visual survey [21,22]. OSR plant species may remain asymptomatic when infected; therefore, appropriate diagnostic methods are necessary to identify a viral infection. There are several diagnostic methods for identifying TuYV, e.g., ELISA (enzyme-linked immuno assay) test [23] and molecular techniques (riboprobes [24] and RT-PCR [25]).

The extent of TuYV incidence in OSR crops appears to be variable, ranging from less than 10% up to 85% infection (depending on the hybrid/variety, growing region, number of aphids, etc.), which affects the crop yield and quality parameters [15,17]. Based on the field data so far, the effect of TuYV on OSR yield varies over a wide range. Graichen and Schliephake [26] demonstrated that experimental plots of OSR yielded approximately 10% less seed (with 100% TuYV infection, based on the ELISA test), while Jones et al. [27] showed that the yield loss may be as high as 46% (with 96% TuYV infection, based on ELISA test). Coutts et al. [28] established that for every 1% increase in the level of TuYV in an OSR crop, there was a 6–12 kg/ha yield reduction. TuYV infection significantly increased individual seed weight (up to 11%); nonetheless, fewer seeds were formed per siliqua [27,28]. The yield loss depends on the OSR phenological stage of TuYV infection. If the crop becomes infected at the end of its growing season, the spread of the virus is much less, and yield and quality losses are minimal [28].TuYV infection diminishes up to 3% of the oil content of seeds but increases up to 44% of erucic acid content; up to 6–11% increase in seed protein content was also recorded [27], while the results by Graichen and Schliephake [26] indicated that TuYV infection did not cause a decrease in the oil and protein contents.

During the IPM of TuYV, it is necessary to implement the eight principles of Integrated Pest Management (IPM) [29]. One of the key agrotechnical factors to be considered is the choice of field location, which is influenced by both the weed and crop species associated with the virus–host plants, as well as the neighbouring crops [5]. Early sowing and dense crop stand (high humidity) may lead to the proliferation of aphids [30,31], while the low germination density of OSR becomes overgrown by weeds, which may serve as virus reservoirs (e.g., *A. retroflexus, C. bursa–pastoris, D. sophia, R. raphanistrum, S. alba*) [5,32]. Plant nutrition deserves a priority for disease management because the application of nitrogen leads to increased aphid population growth, and for this reason, a split application of nitrogen is recommended [33]. The selection of resistant crop hybrids plays an important role in managing TuYV infection. There is only one source of TuYV resistance in the current commercial OSR hybrids, derived from the resynthesised *B. napus* line 'R54', which is linked to a single dominant quantitative trait locus (QTL) on chromosome A04 [34,35]. OSR hybrids' resistance to TuYV is primarily determined by a dominant resistance gene,



which is influenced by environmental factors (temperature) [34]; in addition, resistance-breaking isolates of TuYV increase the selection pressure [35]. The chemical control of viral vectors plays a key role in orchestrating defence against TuYV [15]. Until the withdrawal of neonicotinoids from the EU market, seeds were treated with those active compounds as a preventive measure, which suppressed the early infection of aphids (e.g., clothianidin [36], imidacloprid [27]).

The current study aimed to determine the susceptibility of four commercial hybrids of OSR to TuYV and to determine the effect of the virus on phenotypic (plant height, root neck diameter, number of branches, number of siliquae, siliqua length, number of seeds) and yield parameters (yield, protein content, moisture content, and oil content) in Hungary under field conditions.

## 2. Materials and Methods

The study was conducted during the 2020/21 and the 2021/22 growing seasons in an OSR field near Hegyfalu (47.3528653°; 16.8827556°), a typical field crop growing area in Western Hungary. In those fields where OSR was sown, the agronomic practices were similar every year and followed the usual cultivation practices of the region.

Tests were conducted on four commercial hybrids: Bluestar (Syngenta, Hungary) (without TuYV resistance), PT275 (Corteva, Hungary) (without TuYV resistance), Umberto (KWS, Hungary) (without TuYV resistance), and Architect (Limagrain, Hungary) (with TuYV resistance). These 4 OSR hybrids were planted side by side, each hybrid on 6 m wide and 250 m plot (=1500 m$^2$) in each study year with 2.5 kg seed/ha planting density and 15 cm row spacing. Plots were separated by a 0.4 m uncropped bare soil alley. For each hybrid (6 m × 250 m per hybrid), 6 observation/sampling points were established, and at each point, 5 consecutive plants were sampled/phenotypic parameters measured and grains harvested. Therefore, for each hybrid, 30 plants were included in our studies.

Symptom observation was carried out once per growing season, at the time of leaf sample collection, i.e., at the elongation stage (BBCH 30–39). As with leaf sample collection, observation and sampling were carried out on each plot, at 6 points per plot, with 5 plants per point (i.e., 30 plants per hybrid)). Leaf samples were collected from each plant (2–3 leaves per plant) and sent to the laboratory, where the virus infection was determined by ELISA test. The ELISA test was carried out by the National Food Chain Safety Office (NÉBIH) in Hungary using EPPO PM 7/125 and ELISA kit Loewe 07009S/500.

For the ELISA test, 0.2 g of OSR leaves were used and pressed in 2 mL of homogenising buffer using an electric homogeniser. After sensitisation of the ELISA plate, unbound IgG (anti body) molecules were removed with washing buffer and the tissue fluids of the plant samples, while negative and positive control solutions were added to the ELISA test plate wells. If the sample contained the antigen (TuYV), the virus would bind to the IgG bound to the plate. The IgG labelled with the alkaline phosphatase enzyme is then loaded into the wells of the ELISA plate. If the virus is present (bound to the immobilised IgG) in the well, the enzyme-labelled IgG (conjugate) binds to the other half of the virus. Finally, a solution of the alkaline phosphatase enzyme substrate (para-nitrophenyl phosphate) is loaded into the wells of the ELISA plate. The para-nitrophenyl phosphate compound and the solution prepared from it are colourless. The alkaline phosphatase enzyme starts to cleave the phosphate groups from the substrate molecules, resulting in the yellow-coloured para-nitrophenol. The intensity of the colour change is directly proportional to the concentration of the virus.

The phenotypic parameters were evaluated at maturity (BBCH 80–89). The height was measured from the soil surface to the uppermost branching of the stem, and the root neck diameter was measured (in cm) using a Vernier calliper. After manual harvesting, the number of branches and the siliquae of the plants were recorded. From each plant, 30 siliquae were randomly selected, and the seeds were counted. The siliquae of the OSR plants were subsequently reaped and threshed with a Minibatt sample harvester, and the yield per plant was measured on an analytical balance. The yield per plant was determined

at a standard 6% moisture content. The seed oil, protein, and moisture content were determined in the laboratory using an INSTALAB 600.

The weather conditions were similar during the growing seasons. In 2022, March was found to be colder compared to the 2021 weather data, while May had a higher monthly average temperature than in 2021 (Figure 1).

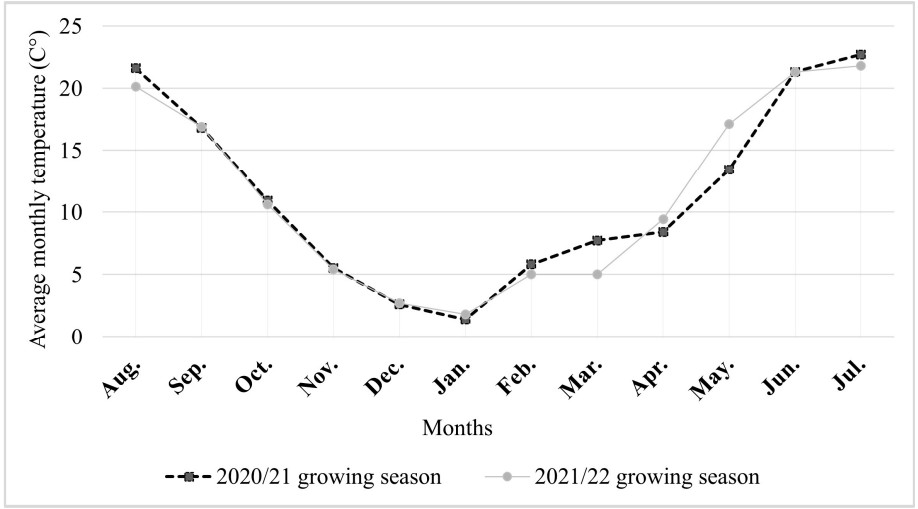

**Figure 1.** Average monthly temperature in the 2020/21 and 2021/22 growing seasons [37].

An analysis of variance (ANOVA) was performed to compare the effect of TuYV on different OSR hybrid parameters. The comparisons were based on TuYV-infected and not-infected plants (based on the ELISA test), which came from plots of each hybrid, and for these, ANOVA and Tukey's pairwise comparisons were used. The $p$-values that are 5% or less were considered to be statistically significantly different [38].

## 3. Results

From the visual survey (Figure 2), more pronounced visual symptoms of anthocyanin discolouration of leaves were observed on plants in the 2020/21 growing season than in the 2021/22 season.

Furthermore, while plants with anthocyanin symptoms were found in all four hybrids in the first growing season, in the second growing season, only Bluestar and Umberto hybrids showed anthocyanin discolouration symptoms. In the first growing season, the infection rates based on symptoms for Umberto and Bluestar hybrids were 56% and 52%, respectively, and 24% for PT275 and Architect (Table 1).

During the second growing season, only the Bluestar hybrid had the highest incidence of symptoms, with 23% and Umberto 10%. No symptoms were detected in PT275 and Architect hybrids (Table 1).

Based on the ELISA test, in the 2020/21 growing season, virus infection was detected in the Bluestar hybrid only, while in the 2021/22 growing season, infected plants were found in all four hybrids. In both growing seasons, the highest infection rate was found in the Bluestar hybrid, with 32% in the first and 90% in the second growing season (Table 1). While PT275, Umberto and Architect hybrids displayed no infections in the 2020/21 season, by contrast, in the 2021/22 season, infection rates for these hybrids were 70%, 36%, and 50%, respectively (Table 1).

Regarding virus-infected plants displaying symptoms appearance confirmed by the ELISA test, 71% of the Bluestar hybrid plants showed virus symptoms in the 2020/21 growing season, with no symptoms detected in PT275, Umberto, and Architect hybrids (Table 1). In the 2021/22 growing season, Bluestar and Umberto hybrids had infection rates of 26% and 9%, respectively, with PT275 and Architect hybrids showing no virus symptoms (Table 1).

**Table 1.** The extent of infection of the tested hybrids based on the visual symptoms and the ELISA test, and the symptom appearance of the plants confirmed by the ELISA test.

| Growing Season | Hybrid | Infection Based on Symptoms (%) | Infection Based on ELISA Test (%) | Symptom Appearance of Infected Plants Confirmed by ELISA Test (%) |
|---|---|---|---|---|
| 2020/21 | Bluestar | 52.00% | 32.00% | 71.43% |
| | PT275 | 24.00% | 0.00% | 0.00% |
| | Umberto | 56.00% | 0.00% | 0.00% |
| | Architect | 24.00% | 0.00% | 0.00% |
| 2021/22 | Bluestar | 23.33% | 90.00% | 25.93% |
| | PT275 | 0.00% | 70.00% | 0.00% |
| | Umberto | 10.00% | 36.67% | 9.09% |
| | Architect | 0.00% | 50.00% | 0.00% |

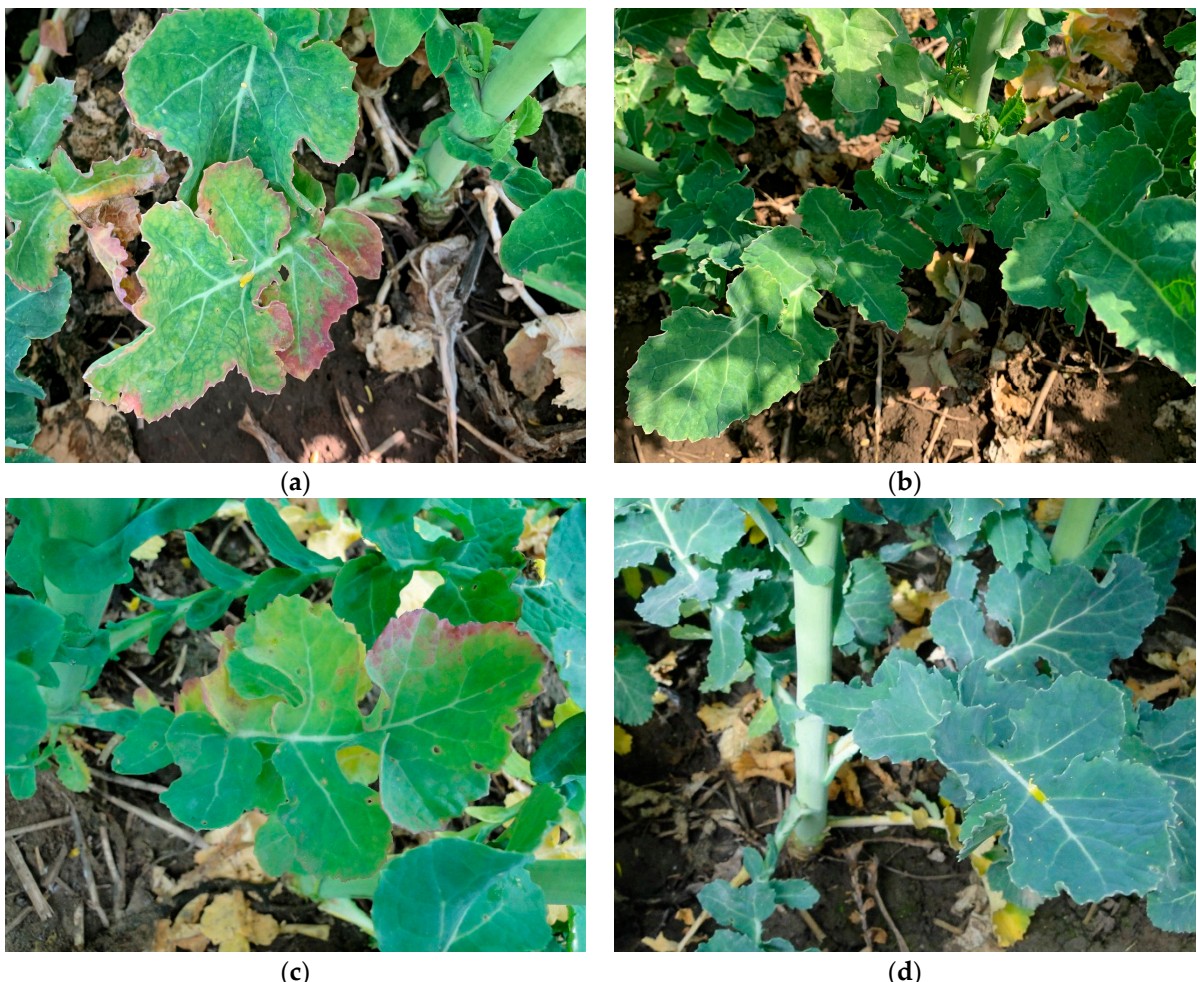

(**a**)　　　　　　　　　　　　　　　　　　　　(**b**)

(**c**)　　　　　　　　　　　　　　　　　　　　(**d**)

**Figure 2.** (**a**) Symptom of a turnip yellows virus (TuYV)-infected winter oilseed rape (OSR) plant and (**b**) OSR plant not infected with TuYV in 2020/21 (**a,b**) and 2021/22 (**c,d**) growing seasons.

Furthermore, some plants showed virus-like symptoms but were not TuYV-infected based on the ELISA test. In the 2020/21 growing season, 28% of plants showed virus-like symptoms in the Bluestar hybrid, and in the 2021/22 growing season, 74% of plants in the

Bluestar hybrid and 91% in the Umberto hybrid were not detected by ELISA test for TuYV (Table 1).

The root neck diameter of TuYV-infected plants was lower in the 2021/22 growing season when compared to the root neck diameter of non-infected plants. This reduction is 0.27 cm (15% reduction) for the Bluestar hybrid, 0.3 cm (19% reduction) for the PT275 hybrid and 0.08 cm (6% reduction) for the Umberto hybrid. However, in the first year, the Bluestar hybrid and in the second year, the Architect hybrid TuYV-infected plants had larger root neck diameters (Figure 3).

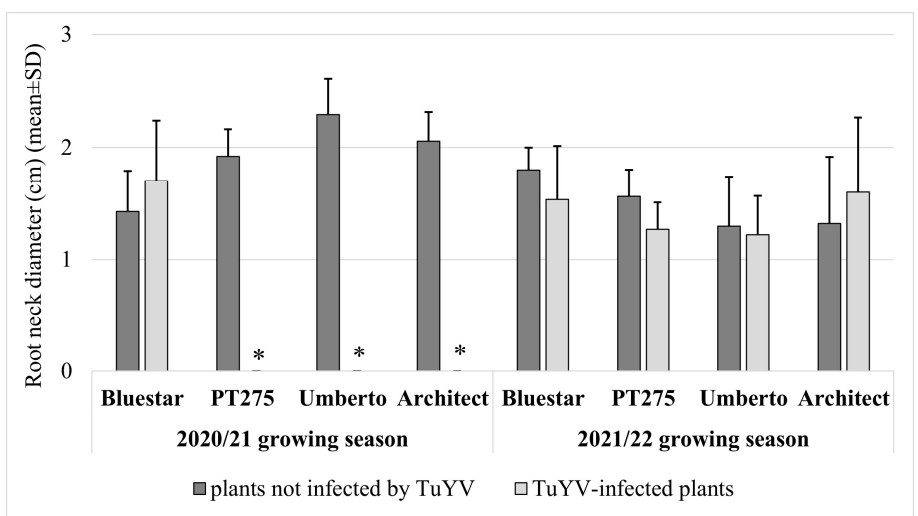

**Figure 3.** The effect of TuYV on root neck diameter in the 2020/21 and 2021/22 growing seasons for the tested hybrids. * No data presented, as no infection was detected in these hybrids.

The average height of infected plants decreased in the second growing season in the Bluestar hybrid (average of 7%) and the PT275 hybrid (average of 11%). The virus-infected Bluestar plants in the first year and the virus-infected Umberto and Architect plants in the second year were taller than the uninfected plants by an average of 3 to 11%, depending on the hybrid (Figure 4).

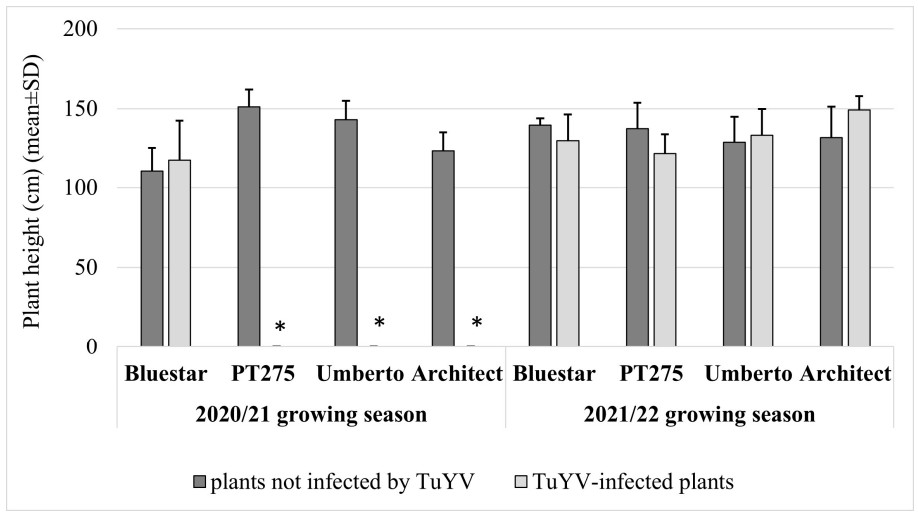

**Figure 4.** The effect of TuYV on plant height in the 2020/21 and 2021/22 growing seasons for the tested hybrids. * No data presented, as no infection was detected in these hybrids.

In the case of yield components, there was no clear effect of the TuYV infection, as in the first year for the Bluestar hybrid and in the second year for the Architect hybrid,

the average number of branches and the number of siliquae per plant were higher on the virus-infected plants. Conversely, regarding the parameters mentioned above, higher values were obtained in the second year of the Bluestar hybrid, the PT275 hybrid, and the Umberto hybrid on uninfected plants.

The largest reduction in the number of branches due to TuYV infection was observed in the second year for the Bluestar hybrid (17%) (Figure 5).

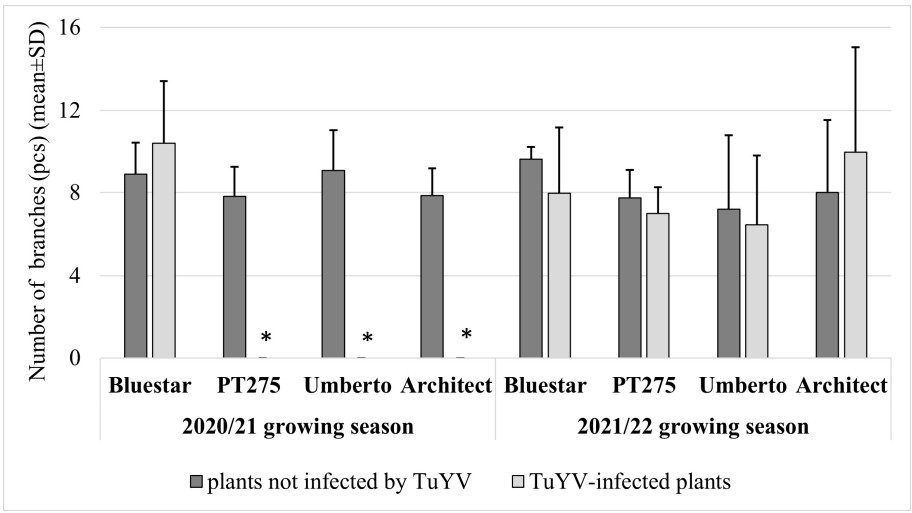

**Figure 5.** The effect of TuYV on the number of branches in the 2020/21 and 2021/22 growing seasons for the tested hybrids. * No data presented, as no infection was detected in these hybrids.

The largest reduction in the number of siliquae was observed for the PT275 virus-infected hybrid in the 2021/22 growing season (43%) (Figure 6).

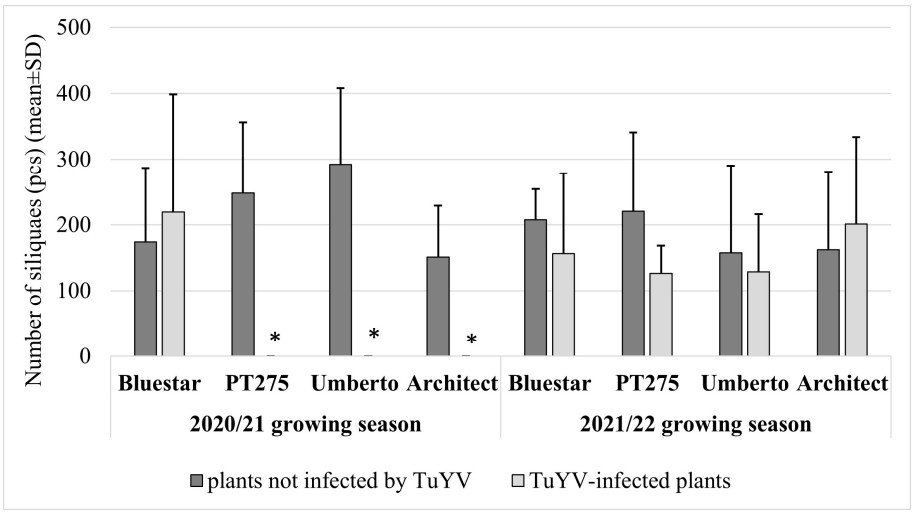

**Figure 6.** The effect of TuYV on the number of siliquae in the 2020/21 and 2021/22 growing seasons for the tested hybrids. * No data presented, as no infection was detected in these hybrids.

The Umberto hybrid showed an average of approximately 2% increase in siliqua length on TuYV-infected plants, while the other hybrids showed a reduction in siliqua length (between <1% and 10%) as a result of infection (Figure 7).

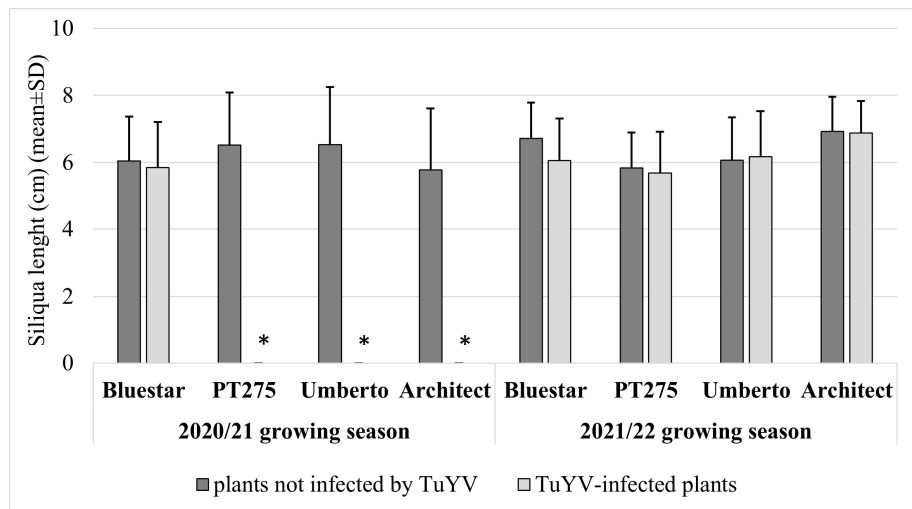

**Figure 7.** The effect of TuYV on siliquae length in the 2020/21 and 2021/22 growing seasons for the tested hybrids.* No data presented, as no infection was detected in these hybrids.

Even though most hybrids (except Umberto and Architect hybrid) had fewer seeds per siliqua on TuYV-infected plants, only Bluestar hybrid (both growing seasons) (4% and 11%) and PT275 hybrid (10%) had a decrease in the number of seeds per siliqua on infected plants. In addition, the Umberto and Architect hybrids showed an increase (4% and 6%) in the number of seeds per siliqua compared to non-infected plants (Figure 8).

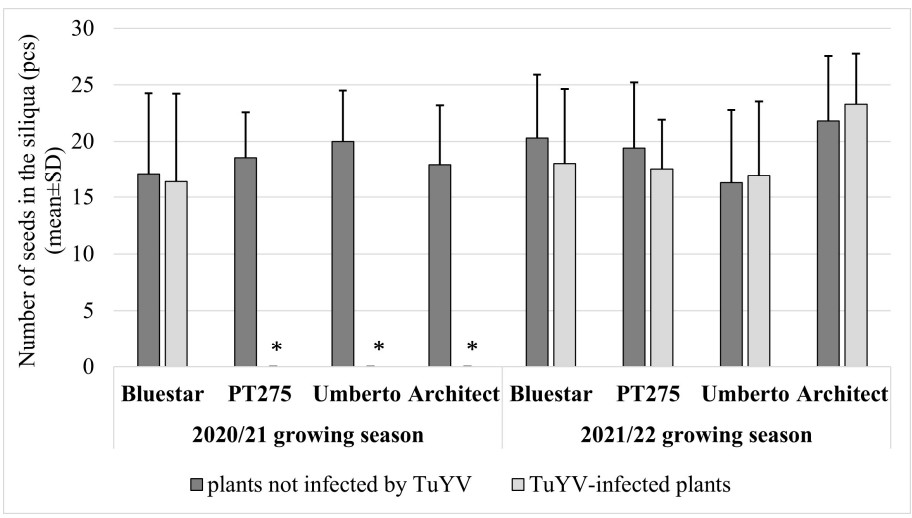

**Figure 8.** The effect of TuYV on the number of seeds in siliqua in the 2020/21 and 2021/22 growing seasons for the tested hybrids. * No data presented, as no infection was detected in these hybrids.

In the second year, the non-infected Bluestar, PT275, and Umberto hybrids delivered higher yields. In contrast, the Bluestar hybrid in the first year and the Architect hybrid in the second year had lower yields in non-infected plants (Figure 9).

This may be explained by the fact that yields were influenced by the number of branches, the number of siliquae, the length of the siliquae, and the number of seeds.

When seed quality parameters (including seed moisture content, protein content, and oil content) were examined in both infected and non-infected OSR crops, no significant differences between TuYV-infected and non-infected plants were obtained for any of the parameters (root neck diameter, plant height, number of branches, number of siliquae, siliqua length, number of seeds, yield per plant, moisture content, protein content, and oil content) and hybrids tested.

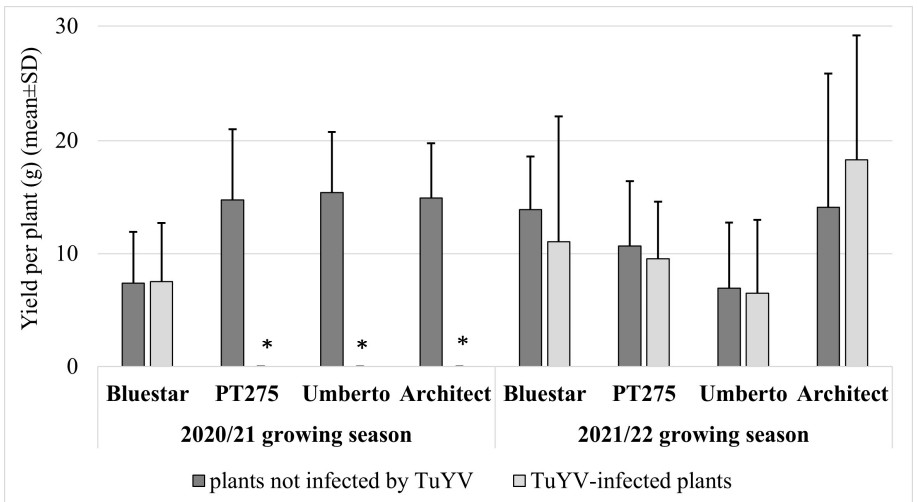

**Figure 9.** The effect of TuYV on yield per plant in the 2020/21 and 2021/22 growing seasons for the tested hybrids. * No data presented, as no infection was detected in these hybrids.

## 4. Discussion

The experiments undertaken during both growing seasons (2020/21 and 2021/2022) have shown that anthocyanin leaf discolouration at the stem elongation stage can be observed in many OSR plants. However, the extent of this discolouration is highly dependent on the season and the hybrid itself. Coutts et al. [28] obtained similar results in Australia, where they found significant differences in virus expression and infection levels when comparing different varieties in their experiments. For hybrids that showed no or only a low percentage of symptoms in the 2021/2022 growing season (PT275, Umberto, Architect), the findings that there were no effects on the phenotypic markers could be related to that or to the importance of additional factors that could be associated with severe disease symptoms.

The current study demonstrated that the degree of discolouration on the OSR leaves does not always correlate with the infection by TuYV. In the case of Architect and PT275 hybrids, no virus symptoms were seen in the 2021/22 growing seasons, but a significant proportion of plants were infected with the virus (ELISA test), while in the other year, plants showed symptoms of likely TuYV infection, but no infection was detected by the ELISA test. TuYV infection levels, as measured, ranged from 0 to 90% in OSR (depending on the year and hybrid). This variation agrees with similar results obtained elsewhere (0–100% in England [20] and 2–100% in Germany [26]). The level of infection by TuYV varied significantly among hybrids, with the Bluestar hybrid showing the highest level of infection. Moreover, infected plants were also found in TuYV-resistant Architect hybrid, which may be due to the genetic diversity of the virus [34,35]. Newbert [8] reported a similar case with the TuYV-resistant Amalie, with a 7% virus infection rate recorded.

With the exception of seed moisture, protein, and oil content, the phenotypic and yield (seed production) parameters of different OSR hybrids displayed variability depending on the season and the virus infection level, but no significant differences between the parameters of TuYV-infected hybrids and non-infected plants were found based on the results of a growing season. These results are in agreement with those of Graichen and Schliephake [26], as TuYV infection had no effect on oil and protein content. The results also demonstrate that the TuYV virus did not affect the phenotypic and yield parameters of OSR in any year or hybrid.

These findings contrast with those of several other studies reporting that TuYV negatively affects plant height, seed number, siliqua length [17], oil content [27], and yield [26,27,39] and positively affects protein content [27].

## 5. Conclusions

TuYV had no overall significant detectable effect on OSR phenotypic and yield parameters in any of the *Brassica napus* L. hybrids tested. There were differences in TuYV infection (based on the ELISA test) between hybrids and even between the growing seasons due to the different seasons and the different susceptibility of hybrids to the virus. Furthermore, there was a difference in infection levels between hybrids based on visual observation; thus, virus infection based on the ELISA test is not consistent with visual detection. For this reason, the decision to intervene or the monitoring of the efficacy of the intervention may be negatively influenced by various factors, including nutrient deficiencies, soil compaction, and low winter temperature. The principles and practices of IPM are likely to reduce the level of virus infection, and in this case, the impact of TuYV is minimal, and the impact of the virus on yield and quality is not significant. For successful IPM implementation, the choice of TuYV-resistant hybrids/varieties is crucial. Although planting resistant hybrids/varieties does not guarantee a virus-free stand due to virus transmission in the field, they can, however, reduce the virus infection rate. In addition to the above preventive measures, the monitoring of aphid infestation, the development of aphid colonies in the flock, and visual identification of symptoms of the virus are alerting signs for further confirmation by the ELISA test.

**Author Contributions:** Conceptualisation, R.V., J.K. and Z.P.; methodology, R.V. and Z.P.; formal analysis, G.T., N.D., J.K., Z.P. and R.V.; data curation, R.V., N.D. and Z.P.; writing—original draft preparation, R.V. and Z.P.; writing—review and editing, G.T., N.D. and J.K. All authors have read and agreed to the published version of the manuscript.

**Funding:** This research received no external funding.

**Data Availability Statement:** The data presented in this study are available on request from the corresponding author.

**Acknowledgments:** The authors would like to thank Patrick L O'Brien and Katalin Körösi for their valuable comments and recommendations.

**Conflicts of Interest:** The authors declare no conflict of interest.

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
