# Peer review of "Turnip Yellows Virus Field Infection in Oilseed Rape: Does It Impact the Yield and Quality?"

_agronomy, doi:10.3390/agronomy13092404_

Round 1
Reviewer 1 Report
My opinion on the reviewed manuscript entitled "Turnip yellows virus field infection in oilseed rape: Does it impact the yield and quality?" is complicated.
The research on the turnip yellows virus (TuYV) is important and interesting, but the research methodology is questionable. Wouldn't it be reasonable to apply an additional factor and assess the degree of infection after virus application to the plant. It is very uncertain to present and discuss results that have not been collected in the field experiment. The artificial introduction of the virus would also make it possible to determine the resistance of varieties. For example the Architekt variety, which is resistant to TuYV, in season 2021/2022, was more severely affected than Umberto, which has not resistant.
Paragraph from line 117 to 125 requires clarification and additional information. Varieties were sown in strips 6 m wide and 250 m long. What constituted a border plot also 6 m wide. The border was along plot? Whether rapeseed and the same variety or a different plant consist border? The border plot was on both sides of the sown variety?
The information provided in manuscript indicated that 5 rapeseed plants were marked (I understood on the 6 x 250 m strip). Why in the sentence (line 122) it is stated that they were 40 cm apart from the other variety. The information is confused.
Line 123 i given that the Authors make 6 observation. This observation was related to the occurrence of TuYV or totally 6 observation during vegetation season was done? In what phases the observations were made or at what time intervals?
Line 125 information is incorrect. The observations were carried out on the same 5 plants (instead of 30) on 6 dates. A total of 30 visual observations were made.
Lines 127-129 Only once leaf sample was collected? If yes in which growing stage samples was collected.
Basic on previous information in manuscript that the observations were carried out on 5 plants of each variety. It was also not stated that leaves for the ELIZA test were taken from a different number of plants. Please explain why the percentage of infection given on the basis of the ELIZA test does not result from the divisor 5? If a different number of tests was performed, it should be supplemented in the methodology.
Figure 3-9 contain incomplete data due to flawed methodology, which in my opinion is a major shortcoming of this manuscript. Impact cannot be assessed without data collection.
Author Response
Lásd a mellékletet.

Reviewer 2 Report
Comments to the Author
- Line 35, missing a punctuation marks ‘)’.
- Line 284, missing a punctuation marks ‘.’.
- Line 161, “TuY-infected” should be changed to “TuYV-infected”.
- Figure 2: According to the manuscript, more pronounced visual symptoms of anthocyanin discolouration of leaves was observed on plants in the 2020/21 growing season than in the 2021/22 season, so the comparison of infection pictures in the growing season of 2021/22 and 2020/21 should be supplemented.
- Line 217-22-: In the second year, the results of Umberto hybrids in Figure 4 are inconsistent with those described in the manuscript.
- In the second year, the non-infected Umberto hybrids delivered higher yields too (Figure 9), but it was not mentioned in this paragraph.
- Is there a significant difference between infected and uninfected data in the column chart?
- The picture is not clear. Please provide clearer picture of the results.
- It would be better if author could supplement the data of one more growing season.
Minor editing of English language required
Author Response
Lásd a mellékletet.

Reviewer 3 Report
The manuscript by Vizi et al. entitled: " Turnip yellows virus field infection in oilseed rape: Does it impact the yield and quality?" studied the susceptibility of Brassica napus L., winter oilseed rape (OSR) hybrid plants to infection by turnip yellows virus (TuYV). Three susceptible hybrids and one resistant hybrid were tested during two growing seasons 2020/2021 and 2021/2022. The authors measured the virus content using Sandwich ELISA and measured virus effects on various phenotypic parameters and yield. The author conclude that visual symptoms were not always indicative of the virus (based on ELISA) and no other negative effects were detected when testing yield and various phenotypic markers.
Comments for the authors:
TuYV is known to cause damage to various crops including Brassica and it is very important to make a conclusion regarding the phenotypic effects of the virus. Because symptomatic plants of three of the hybrids grown in the 1st growing season were negative for TuYV by ELISA, the authors did not include these hybrids in their analyses and actually their conclusions are limited to one growing season only. This should be mentioned in the Abstract and inside the MS. In addition, any conclusion regarding symptoms that are not related to virus presence cannot be made unless RT-PCR is done to confirm the presence or absence of the virus.
The presented results are of the hybrids that showed no or low percent of disease symptoms at the 2nd growing season to begin with, so the findings that there were no effects on the phenotypic markers could be related to that or to the importance of additional factors that could be associated with severe disease symptoms. The authors should make this reservation to their results. Since the same hybrids were symptomatic at the 1st growing season this could emphasize the seasonal importance of the phenotypic markers' study especially if RT-PCR was used to detect the virus.
The authors need to tone down their conclusions by emphasizing those reservations.
Specific comments:
Line 34: use abbreviation for 2nd time use of Alternaria-A.
Line 94:add a virus host reservoir
Line 100: hybrids'
Lines 139: and 145 because line 139 is correct and the APS conugant antibodies bind the virus the color intensity is directly correlated to the virus. Not indirectly.
Line 161: TuYV
Line 218: figure 4 Umberto was not taller than control.
Line 246: "(except Umberto hybrid)" also Architect
Line 276: please be specific regarding the years.
Lines 289-292: not clear
Line 314: "they can however reduce the virus infection rate"-based on what?
Author Response
Lásd a mellékletet.

Round 2
Reviewer 1 Report
Authors response for most of the comments and improve manuscript. My opinion in conclusion additional correction before publishing is needed:
The Authors didn't try to answer on the question about plant resistance and their reaction in the field condition. Question from last review: "For example the Architekt variety, which is resistant to TuYV, in season 2021/2022, was more severely affected than Umberto, which has not resistant".
Author Response
Lásd a mellékletet.
